# Impact of *hfq* and *sig*E on the tolerance of *Zymomonas mobilis* ZM4 to furfural and acetic acid stresses

Hoda Nouri[1], Hamid Moghimi[1]*, Sayed-Amir Marashi[2], Elahe Elahi[3]

**1** Department of Microbial Biotechnology, School of Biology, College of Science, University of Tehran, Tehran, Iran, **2** Department of Biotechnology, College of Science, University of Tehran, Tehran, Iran, **3** School of Biology, College of Science, University of Tehran, Tehran, Iran

* hmoghimi@ut.ac.ir

**Data Availability Statement:** All relevant data are within the manuscript and its Supporting Information files.

**Funding:** The author(s) received no specific funding for this work.

## Abstract

*Zymomonas mobilis*, as an ethanologenic microorganism with many desirable industrial features, faces crucial obstacles in the lignocellulosic ethanol production process. A significant hindrance occurs during the pretreatment procedure that not only produces fermentable sugars but also releases severe toxic compounds. As diverse parts of regulation networks are involved in different aspects of complicated tolerance to inhibitors, we developed ZM4-hfq and ZM4-sigE strains, in which *hfq* and *sig*E genes were overexpressed, respectively. ZM4-hfq is a transcription regulator and ZM4-sigE is a transcription factor that are involved in multiple stress responses. In the present work, by overexpressing these two genes, we evaluated their impact on the *Z. mobilis* tolerance to furfural, acetic acid, and sugarcane bagasse hydrolysates. Both recombinant strains showed increased growth rates and ethanol production levels compared to the parental strain. Under a high concentration of furfural, the growth rate of ZM4-hfq was more inhibited compared to ZM4-sigE. More precisely, fermentation performance of ZM4-hfq revealed that the yield of ethanol production was less than that of ZM4-sigE, because more unused sugar had remained in the medium. In the case of acetic acid, ZM4-sigE was the superior strain and produced four and two-fold more ethanol compared to the parental strain and ZM4-hfq, respectively. Comparison of inhibitor tolerance between single and multiple toxic inhibitors in the fermentation of sugarcane bagasse hydrolysate by ZM4-sigE strain showed similar results. In addition, ethanol production performance was considerably higher in ZM4-sigE as well. Finally, the results of the qPCR analysis suggested that under both furfural and acetic acid treatment experiments, overproduction of both *hfq* and *sig*E improves the *Z. mobilis* tolerance and its ethanol production capability. Overall, our study showed the vital role of the regulatory elements to overcome the obstacles in lignocellulosic biomass-derived ethanol and provide a platform for further improvement by directed evolution or systems metabolic engineering tools.

**Competing interests:** The authors have declared
that no competing interests exist.

## Introduction

To decrease reliance on limited fossil reserves and to control global warming, an alternative
environment-friendly fuel is important. Bioethanol, as a sustainable energy source, can be a
substitution for the finite resources of fossil fuels and is considered to decrease the rate of envi-
ronmental pollution [1]. In the process of bioethanol production, substrate usage and product
formation capabilities of microorganisms play central roles. *Zymomonas mobilis* has remark-
able industrial characteristics because of its notable usage of the Entner-Doudoroff (ED) path-
way [2]. Various enzymes within the Embden–Meyerhof–Parnas (EMP) pathway, Krebs cycle
and pentose phosphate pathway (PPP) are not identified in this microorganism. These trun-
cated pathways and enzyme deficiencies drive more carbon flux into the ED pathway and
bioethanol production [1, 2]. Compared to other ethanol producing microorganisms, *Z. mobi-
lis* shows a higher sugar uptake rate, much higher ethanol tolerance (up to 16% v/v), higher
yield of ethanol production, and a border pH range (3.5–7.5) of ethanol production [2–4]. Fur-
thermore, fermentation by *Z. mobilis* does not need aeration control, and consequently,
reduces the bioethanol production costs [2–4].

Lignocellulosic wastes can be turned into suitable sugars for bioethanol production. In the
conversion process for releasing fermentable sugars, some toxic compounds are released that
may result in process failure by causing the microorganism's viability to collapse and/or reduc-
ing the yield of ethanol production [5]. Acetic acid formation from acetyl groups in hemicellu-
loses causes a reduction in cellular pH and acetate accumulation. Furfural is generated in the
hydrolysis process through acidic treatment at high temperatures from xylose, which is a con-
stituent of hemicelluloses [1, 5]. These compounds are harmful to the activity of certain essen-
tial enzymes and cause detrimental effects on cellular function. Consequently, detailed
researches have been undertaken to clarify the role of different parts of cellular responses to
inhibitors [1]. Transcriptome profiling in the presence of furfural, acetic acid, ethanol and phe-
nolic aldehyde has been previously analyzed. The results suggested that transcriptional control
and regulatory elements play crucial roles in stress responses of cellular metabolism [3, 6–9].
Furthermore, it was shown that central carbon metabolism, membrane biogenesis and general
stress response proteins are involved in tolerance to lignocellulosic inhibitors in *Z. mobilis* [1,
3]. To alleviate the negative effects of these inhibitors, many efforts have been undertaken,
including mutagenesis [10], detoxification [7], adaptive evolution [11], genome shuffling [12]
and systems biology-driven approaches [4, 13]. Improving the tolerance of *Z. mobilis* to inhibi-
tors proposed that it is associated with some regulatory systems that altered in stressful envi-
ronments [11]. These findings, together with the transcriptome analysis, have confirmed the
critical role of regulatory and sigma factors in *Z. mobilis* cellular responses to various stresses.
In this context, many efforts are focused on engineering the transcriptional regulators and
transcription machinery, small RNAs and RNA chaperones [14–18].

Hfq, as a global regulator, contributes to stress response in *Z. mobilis*. Moreover, the homo-
log of Hfq in *Saccharomyces cerevisiae* (Lsm protein) confirmed a similar function in exposure
to inhibitors [18, 19]. Yang *et al*. disrupted *hfq* in *Z. mobilis* and the mutant strain showed
more sensitivity to various inhibitors. Besides, they overexpressed Lsm protein (Hfq homo-
logue) in *S. cerevisiae* and displayed improved acetate and 5-(hydroxymethyl) furfural resis-
tances compared to the control [18]. In a similar study, Cho *et al*. (2017) used untranslated
regions (5′TTRs) as regulatory elements to encode the RNA-binding protein Hfq. Their results
showed down-regulation of ethanol stress-related genes [19].

RpoE (sigE) is a transcription factor in *Z. mobilis* and its role was transcriptionally approved
in response to ethanol stress [20]. Seo *et al*. proposed that rpoE (sigma E) in *Z. mobilis* is a
RpoS-like regulatory factor and plays a critical role in various stress conditions [20]. Recently,

Benoliel *et al*. (2019) overexpressed sigma 32 and sigma E in *Z. mobilis* and confirmed that both of them are related to heat shock stress [15].

As we can conclude from previous researches, this study aimed to consider the role of over-production of *hfq* and *sig*E genes in response to acetic acid, furfural and sugarcane bagasse hydrolysate. These results will help us clearly identify the accurate mechanisms of these two genes in furfural and acetic acid stresses and provide a deep insight into metabolic engineering purposes.

# Materials and methods

## 2.1. Bacterial strains, plasmid vectors and culture conditions

*Zymomonas mobilis* ZM4 (ATCC 31821) was used as a parental strain and was grown anaero-bically on RM medium containing the following per liter: 10 g yeast extract, 20 g glucose and 1 g $KH_2PO_4$ at 30˚C. *Escherichia coli* DH5α was used for the construction of the final expression cassette in *Z. mobilis*. pUC19 as a cloning vector and pBBR1MCS-2 as a shuttle vector were used in *E. coli* and *Z. mobilis*, respectively. *E. coli* was prepared in Luria–Bertani (LB) medium supplied with ampicillin or kanamycin (100 μg.mL$^{-1}$) as a selectable marker. Recombinant *Z. mobilis* strains were supplemented with 100 μg.mL$^{-1}$ of kanamycin. For monitoring the growth profile in addition to consumed sugar and ethanol production, anaerobic bottles containing 50 ml of RM medium were used. To assess stress condition, treatments with 1, 2 and 3 gL$^{-1}$ of fur-fural and 2.5 and 5 gL$^{-1}$ acetic acid were used. Also, sugarcane bagasse hydrolysate was applied for analyzing multiple inhibitor effects besides industrial carbon source preparation (S1 Graphical abstract).

## 2.2. Plasmid preparation and construction of recombinant strains

General molecular biology procedures from DNA extraction to transformation in *Z. mobilis* were done according to standard protocols [21]. All PCR reactions were done in a Multi-Gene OptiMax thermocycler (Labnet) by using pfu or Taq DNA polymerases (Bioneer South Korea). DNA purification and gel extraction were performed using an AccuPrep DNA Extraction kit (Bioneer, South Korea). All primers are listed in S1 Table. To overex-press *sigE* (ZMO1404) and *hfq* (ZMO0347) genes in *Z. mobilis* ZM4, recombinant plasmids pBBR1MCS-2/sigE and pBBR1MCS-2/hfq were constructed in *E. coli* DH5α. For this pur-pose, we used 250 bp upstream and 200 bp downstream of the pyruvate decarboxylase gene (pdc CDS) (ZMO1360) in *Z. mobilis* as a strong native promoter and terminator, respec-tively. pdc-F/pdc-R and Tpdc-F/Tpdc-R primer pairs were used to amplify 249 and 200 nucleotide sequence fragments upstream and downstream of the pyruvate decarboxylase gene in *Z. mobilis* DNA, respectively. *hfq* (486 nucleotides) and *sig*E (699 nucleotides) genes were amplified using primer pairs hfq-F/hfq-R and sig-F/sig-R, respectively from *Z. mobilis* DNA. After gel extraction, all amplified fragments were subcloned to pUC19 plasmid in *E. coli* DH5α. Using SOEing PCR techniques, the pdc promoter region was fused in the upstream of the *hfq* and *sig*E fragments. To construct PCR products representing *sig*E cas-sette gene, sig-F-pdc/sig-R′ and pdc-F′/pdc-R-sig pair primers were applied for *sig*E gene and pdc promoter amplification, respectively. After purification, both fragments were joined through SOEing PCR by pdc-F′/sig-R′ primers. A similar approach was applied to fuse 200 bp terminator fragment to 3′ terminus of the final cassette and the 1148 bp final PCR product was subcloned in *E. coli* and after purification, followed by digesting with *Bam*HI and *Eco*RI. Ligation was done into the pBBR1MCS-2 vector and transformed to *E. coli*. The recombinant plasmid pBBR1MCS-2/sigE was purified and sequenced. In the case of *hfq* gene, hfq-R′/hfq-F-pdc and pdc-F/pdc-R-hfq pair primers were involved in the

amplification of *hfq* and promoter genes, respectively. After fragment purification, overlapping based SOEing PCR system by using pdc-F´/hfq-R´ primers resulted in 735 bp PCR products. In the same way, pdc terminator was fused to pdc/hfq fragment and the final 935 bp cassette was used to obtain pBBR1MCS-2/hfq. The recombinant plasmid pBBR1MCS-2/hfq was purified and sequenced. Both of the recombinant plasmids (pBBR1MCS-2/sigE and pBBR1MCS-2/hfq were transformed to the *Z. mobilis* via the electroporation (Gene Pulser Xcell[TM] (Bio-Rad). Briefly, fresh mid-log *Z. mobilis* cells were suspended in 200 μl glycerol and appropriate condition (10 ms in a 0.2 cm cuvette and 2.5 kV) was applied for entering the recombinant plasmids to the cell [22]. After that, 3 mL of RM was added to cells and incubated at 30˚C for 20 h. At the end of incubation, the cells were grown on the RM selective medium containing kanamycin (100 μg.mL$^{-1}$) and incubated at 30˚C for three days. PCR identified the true transformants. Recombinant strains pBBR1MCS-2/sigE and pBBR1MCS-2/hfq were named ZM4-sigE and ZM4-hfq, respectively (S1 Fig).

## 2.3. The growth profile of parental and recombinant strains under furfural and acetic acid stresses

*Z. mobilis* ZM4 and two recombinant strains, ZM4-sigE and ZM4-hfq were cultured in RM at 30˚C to reach an optical density of 0.8. The harvested cells were centrifuged and washed with normal saline in order to prepare a diluted cell with 0.8 absorbance at 600 nm. 10% of seed culture suspension was inoculated in each of the anaerobic bottles containing RM medium plus 1,2 and 3 gL$^{-1}$ furfural or 2.5 and 5 gL$^{-1}$ acetic acid. RM medium without inhibitors was considered as control. All the bottles were incubated at 30˚C for 24 h followed by serial dilution preparation [22]. 5 μl of each dilution sample was spotted on RM plate and then incubated in the previously mentioned condition. Three replications were done in all experiments.

## 2.4. Fermentation profiles under furfural and acetic acid stresses or sugarcane bagasse hydrolysate

An experimental procedure was conducted to access the parental and recombinant strains in the presence of one or multiple inhibitors. As mentioned above, 5 ml of washed and diluted fresh cells were inoculated to 45 ml of RM medium containing furfural (1, 2 and 3 gL$^{-1}$) or acetic acid (2.5 and 5 gL$^{-1}$). In the case of multiplex inhibitors, sugarcane bagasse hydrolysate was substituted with sugar. The sugarcane bagasse hydrolysate was provided by acid hydrolysis method [23] and its composition is presented in S2 Table. During 60 h fermentation period, sampling was done in 6 h regular intervals for evaluating growth rate and consumed sugar, and every 12 hours for produced ethanol. The absorbance value of the cell suspension was analyzed spectrobolometrically at 600 nm (RayLeigh UV1601, China). Glucose oxidase based method was used for quantifying residual sugar content of samples [23]. Ethanol content was measured by HPLC method (Agilent Hiplex H, 300 mm×7.7 mm) in the following condition: 0.05 M sulfuric acid as mobile phase at a flow rate of 0.6 ml/min and a column temperature of 35˚C. A peak area of each sample in comparison to standard controls was used for ethanol quantification. All experiments were performed in triplicate [22].

## 2.5. RNA isolation qRT-PCR

The gene expression of some indicator genes were investigated in recombinant and wild type strains of *Z. mobilis* ZM4 treated with furfural and acetic acid. The wild type and recombinant strains were treated with 3 gL$^{-1}$ furfural or 5 gL$^{-1}$ acetic acid. After 6 h, bacterial cells were collected and total cellular RNA was purified by RNX-PLUS (Sinaclone,

Iran). All RNA samples were diluted in 30 μl of 0.1% DEPC (diethyl pyrocarbonate) water. Reverse transcription (RT) reaction was done by RevertAid RT Reverse Transcription Kit (Thermo Fisher Scientific) following the manufacturer's instructions. The qPCR reactions were performed in the StepOnePlus™ Real-Time PCR Systems (Applied Biosystems). The expression pattern of some indicator genes (*sig*E, *hfq*, *pdc*, *adh*, *xylR* and *nhaA*) was normalized by the *rrsA* gene (16S RNA) as an internal control. All primers are listed in S1 Table. To analyze the gene expression, the changes in gene expression level were considered as fold-change using the ΔΔCt method and standard curves of each gene compared to the normalization gene (*rrsA*). All reactions were repeated in triplicate [24].

## Results and discussion

### 3.1. Effect of stresses on the growth of parental and recombinant strains of *Z. mobilis*

The parental and recombinant strains of *Z. mobilis* were submitted to different concentrations of furfural and acetic acid treatments. The performance of all strains during the cell growth on the inhibitors is displayed in Fig 1. As we expected, when no inhibitor was added to the culture medium, the parental and both recombinant strains showed similar growth pattern. These observations confirmed that the growth ability of ZM4-sigE and ZM4-hfq were not influenced during genetic engineering procedures and remained constant like control sample. In the case of furfural stress, the increase in furfural content has coincided with a decrease in growth. Despite no considerable differences between the two recombinants, the parental strain showed a remarkable reduction in growth under furfural stress. Investigation of acetic acid effect on growth revealed that this inhibitor had a more negative effect on the ZM4-hfq strain compared to ZM4-sigE. The growth of ZM4-sigE was higher than the control and ZM4-hfq under all acetic acid concentrations.

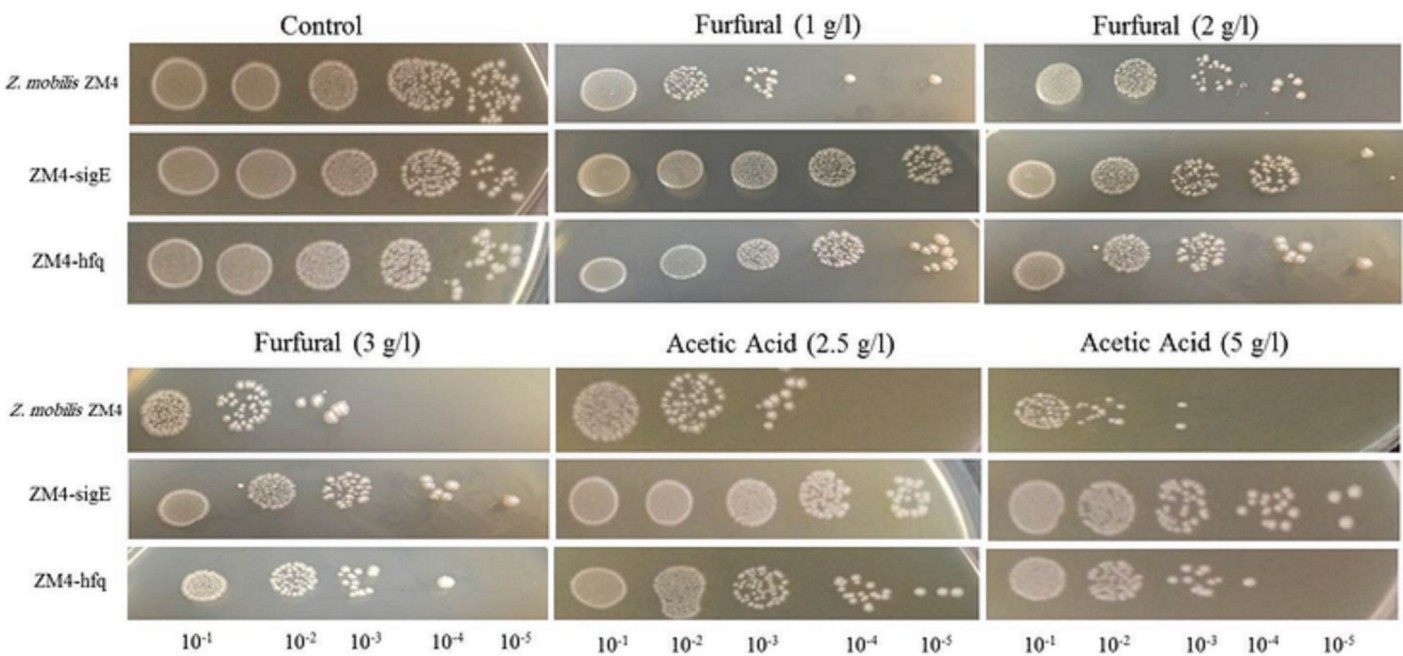

**Fig 1. Growth patterns of the wild type, ZM4-sigE and ZM4-hfq strains in RM medium containing different levels of furfural and acetic acid.**

## 3.2. Effect of furfural and acetic acid on fermentation profiles of *Z. mobilis*, ZM4-sigE and ZM4-hfq

The lignocellulosic inhibitors consist of aldehydes such as HMF and furfural and weak acids, mainly acetic acid [23]. The bacterial cell membrane permeability, lipid arrangement and fluidity, and also the physiological processes, including electron transport chain, nutrient intake and uptake and energy transduction, were affected by ethanol and these inhibitors' toxicity [22]. The tolerance to these inhibitors is a complicated phenotype that is organized by unknown regulatory mechanisms. The development of resistant strains by rational and evolutionary engineering are useful approaches to find genetic elements related to inhibitors' tolerance [2].

To map the influence of different concentrations of furfural and acetic acid on wild type and recombinant strains, fermentation performance and the growth profile were investigated. In the stress-free culture flasks, growth pattern, sugar consumption and ethanol production were unaffected in the three strains mentioned above (Fig 2A). The culture was challenged with increasing concentrations of furfural, from 1 to 3 $gL^{-1}$, to analyze different strains' characteristics.

Either 2 $gL^{-1}$ or 3 $gL^{-1}$ furfural does little inhibition to the growth of ZM4-hfq or ZM4-sigE. However, in medium containing 3 $gL^{-1}$ furfural, wild type growth remarkably reduced. (Fig 2 and Table 1). It appeared that furfural was more toxic to ZM4-hfq cell growth than ZM4-sigE, at a given concentration. In *Z. mobilis* ethanol production is closely associated with cell growth and significantly reduced by inhibitory effects of toxic compounds [3]. Tan *et al.* (2015) study demonstrated that RpoD mutants and wild type strain had no significant difference in media without furfural stress. Although, a drastic difference between mutant and wild type strains was detected when furfural concentration was gradually increased [22]. Yang *et al.* (2010b) designed an *hfq* mutant (AcRIM0347) and demonstrated that the growth rates were lower than the control strain under acetate, vanillin, furfural, and HMF stresses [18]. At 3 $gL^{-1}$ furfural, ethanol production is about three times higher in ZM4-sigE in comparison to the wild type. ZM4-sigE showed maximum ethanol yield of 0.43 $g_p\ g_s^{-1}$ with approximately 12 h retardation compared to non-stress condition (0.5 $g_p\ g_s^{-1}$ after 24 h). Ethanol production and sugar consumption in higher concentrations of furfural demonstrated the highest ability of ZM4-sigE to convert sugar to ethanol and tolerate toxic conditions (Fig 2B, 2C and 2D and Table 1). Consequently, ethanol production also reduced proportionally to the decrease in sugar depletion. Benoliel *et al.* (2019) overexpressed *sig*E in *Z. mobilis* and evaluated the recombinant strain in response to high temperatures and ethanol stresses. According to their results, the overexpression of *sig*E led to significant improvement in high temperature tolerance, but there is no clear evidence that this sigma factor protects *Z. mobilis* in ethanol stress [15].

In culture conditions with 3 $gL^{-1}$ furfural, the growth is considerably retarded in the wild type strain, but both recombinants showed higher values than the control. The wild type and ZM4-hfq strains reached very low ethanol, 20% and 38% of a normal environment, respectively. These findings supported that the fermentation performance and ethanol production are largely inhibited by 3 $gL^{-1}$ furfural in two ZM4-hfq and control strains.

As a result, the ethanol production of ZM4-sigE reached to the highest content after 36 h (5.8 $gL^{-1}$), which was significantly higher than ZM4-hfq (3.8 $gL^{-1}$) and parental strain (2 $gL^{-1}$). Besides, our results showed that the furfural stress adversely affects the bacterial growth rate (Fig 2). In the culture medium with 3 $gL^{-1}$ furfural, ZM4-sigE reached the highest cell density of about 1.2 ($OD_{600}$) after 36 h, but the highest monitored growth rate for ZM4-hfq and wild type were 0.7 and 0.4 ($OD_{600}$), respectively. Cho *et al.* (2017) constructed an *hfq* deleted *Z. mobilis* 8b by using homologous recombination technique and evaluated the mutant strain

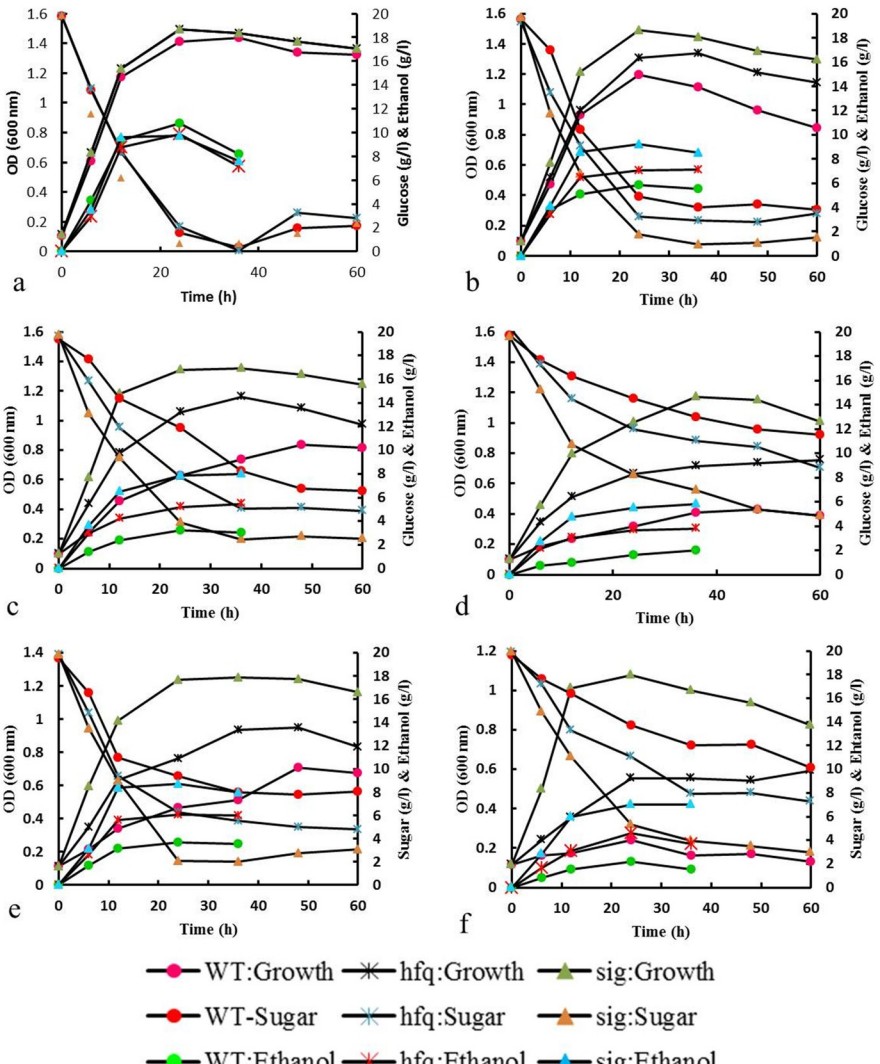

**Fig 2.** Effects of furfural (b:1, c:2 and d: 3 gL⁻¹) and acetic acid (e: 2.5 and f: 5 gL⁻¹) inhibitors on glucose consumption, growth, and ethanol production of ZM4-sigE, ZM4-hfq and wild type. The results are obtained from mean values of triplicate experiments and (a) was run without stress as a control condition.

under 5% ethanol stress. They reported a growth defect in the mutant strain relative to the control strain under ethanol stress [19]. Yang *et al.* (2010b) showed that the *hfq* mutant (AcRIM0347) had a deleterious effect in tolerance to lignocellulosic inhibitors [18]. Our results confirm Yang *et al.* study [18] but the ZM4-sigE strain showed better fermentation performance than ZM4-hfq. Also, ZM4-hfq showed more tolerance to lignocellulosic inhibitors than wild type.

Here, we summarized the ethanol production performances by Z. *mobilis* in the presence of furfural inhibitor (Table 2). In this table, the yield of ethanol shows changes between 32–98% in different strains of Z. *mobilis* [22, 25]. Our recombinant strains were able to reach 78–88% ethanol yield in various furfural and acetic acid concentrations. These results indicated that the ethanol fermentation performance in the presence of furfural was impeded significantly and more efforts are needed to improve Z. *mobilis* when grown in high concentrations of furfural (Table 2).

**Table 1. Fermentation characteristics of the parental and recombinant strains in RM medium containing different concentrations of acetic acid and furfural.**

| | | Wild type | | | ZM4-hfq | | | ZM4-sigE | | |
|---|---|---|---|---|---|---|---|---|---|---|
| | | Ethanol $(g_p\ L^{-1})$ | Yield $(g_p\ g_s^{-1})$ | Productivity $(g_p\ L^{-1}\ h^{-1})$ | Ethanol $(g_p\ L^{-1})$ | Yield $(g_p\ g_s^{-1})$ | Productivity $(g_p\ L^{-1}\ h^{-1})$ | Ethanol $(g_p\ L^{-1})$ | Yield $(g_p\ g_s^{-1})$ | Productivity $(g_p\ L^{-1}\ h^{-1})$ |
| control | 24 | 9 | 0.48 | 0.37 | 8.8 | 0.49 | 0.36 | 9.6 | 0.5 | 0.4 |
| | 36 | 9.8 | 0.49 | 0.27 | 9.8 | 0.5 | 0.27 | 9.8 | 0.51 | 0.27 |
| Acetic acid (2.5 gL$^{-1}$) | 24 | 3.1 | 0.30 | 0.12 | 5.58 | 0.41 | 0.23 | 8.36 | 0.47 | 0.35 |
| | 36 | 3.6 | 0.31 | 0.10 | 6.07 | 0.42 | 0.16 | 8.7 | 0.48 | 0.24 |
| Acetic acid (5 gL$^{-1}$) | 24 | 1.5 | 0.25 | 0.06 | 3.1 | 0.35 | 0.13 | 5.9 | 0.40 | 0.25 |
| | 36 | 2.16 | 0.28 | 0.06 | 4.6 | 0.38 | 0.12 | 7.01 | 0.43 | 0.19 |
| Furfural (1 gL$^{-1}$) | 24 | 5.11 | 0.34 | 0.16 | 6.5 | 0.40 | 0.19 | 8.6 | 0.48 | 0.25 |
| | 36 | 5.82 | 0.37 | 0.21 | 7.05 | 0.42 | 0.27 | 9.2 | 0.49 | 0.35 |
| Furfural (2 gL$^{-1}$) | 24 | 2.4 | 0.32 | 0.08 | 4.22 | 0.35 | 0.14 | 6.5 | 0.41 | 0.21 |
| | 36 | 3.2 | 0.28 | 0.1 | 5.45 | 0.37 | 0.17 | 8 | 0.47 | 0.27 |
| Furfural (3 gL$^{-1}$) | 24 | 1 | 0.19 | 0.04 | 3.03 | 0.37 | 0.12 | 4.73 | 0.41 | 0.2 |
| | 36 | 2 | 0.25 | 0.04 | 3.8 | 0.40 | 0.07 | 5.8 | 0.43 | 0.12 |
| Sugarcane bagasse hydrolysate | 24 | 0.1 | 0.04 | 0.004 | 1.5 | 0.2 | 0.06 | 2 | 0.36 | 0.08 |
| | 36 | 0.22 | 0.14 | 0.006 | 1.5 | 0.21 | 0.04 | 2.3 | 0.36 | 0.06 |

**Table 2. Ethanol fermentation performance of *Z. mobilis* strains in various furfural and acetic acid concentrations.**

| *Z. mobilis* strain | Strain improvement | Technique(s) | Inhibitor concentration | Ethanol $(gL^{-1})$ | Yield | References |
|---|---|---|---|---|---|---|
| ZM4 | ZM4-MF1, 2, 3 Recombinant strain (Sigma factor RpoD) | gTME (error prone PCR) | Furfural (3 gL$^{-1}$) | 9.8 | 98% | [22] |
| ZM4 | ZMF3-3 (Adapted strain to furfural) | Adaptive laboratory evolution (ALE) | Furfural (3 gL$^{-1}$) | 52 | 94.84 | [11] |
| ZM4 | ZMA7-2 (Adapted strain to acetic acid) | ALE | Acetic acid (7 gL$^{-1}$) | 50 | 91.9% | [11] |
| AcR | AcR (Acetate-tolerant strain) | NTG mutagenesis | Sodium acetate (20 gL$^{-1}$) | 51 | 92% | [37] |
| ZM4 | ZM5510-ZM6014 (Acetate-tolerant mutant) | NTG mutagenesis and adaptation | Acetic acid (1.4 or 1.6%) | 20 | 66% | [38] |
| ZM4 | ZM401 (Acetate-tolerant mutant) | NTG mutagenesis | Acetic acid (10.5 gL$^{-1}$) | 27.5 | 46% | [26] |
| AQ8 | 532–533 (Mutant) | Genome shuffling | Acetic acid (5 gL$^{-1}$) and furfural (3 gL$^{-1}$) | 21.49 | 81–82% | [12] |
| ZM4 | ZMA-142 (Acetate-tolerant mutant) | NTG mutagenesis and ALE | Sodium acetate (195 mM) | 40 | 42–64% | [25] |
| ZM4 | ZMA-167 (Acetate-tolerant mutant) | NTG mutagenesis and ALE | Sodium acetate (195 mM) | 40 | 32–42% | [25] |
| AcR | AcR (Acetate-tolerant strain) | - | Sodium acetate (10 gL$^{-1}$) | 8 | 80% | [34] |
| CP4 | F211 and F27 (Mutant) | Error-prone PCR-based whole genome shuffling | Furfural (3 gL$^{-1}$) | 2.6 | 92–94% | [39] |
| AQ8 -AQ9 | PH1-29 (Mutant) | Multiplex atmospheric and room temperature plasma (mARTP) mutagenesis | Acetic acid (5 and 8 gL$^{-1}$) | 22 | 84% | [10] |
| ZM4 | ZM4-hfq (Recombinant strain) | Overexpression of transcriptional regulatory factors | Furfural (3 gL$^{-1}$) | 3.8 | 78% | This work |
| ZM4 | ZM4-hfq (Recombinant strain) | Overexpression of transcriptional regulatory factors | Acetic acid (5 gL$^{-1}$) | 4.6 | 80% | This work |
| ZM4 | ZM4-sigE (Recombinant strain) | Overexpression of transcriptional regulatory factors | Furfural (3 gL$^{-1}$) | 5.8 | 86% | This work |
| ZM4 | ZM4-sigE (Recombinant strain) | Overexpression of transcriptional regulatory factors | Acetic acid (5 gL$^{-1}$) | 7.1 | 88% | This work |

Attempts were made to determine the effects of acetic acid on ethanol fermentation performance. In this case, the parental strain was remarkably affected by acetic acid that was added to the culture medium. Based on the results, when acetic acid increased to 5 $gL^{-1}$, the wild type presented notable less growth (0.24 $OD_{600}$) than those observed in the non-stress situation (1.45 $OD_{600}$). Also, more than half of the sugar remained in the medium of the wild type strain, so very little ethanol (2.1 $gL^{-1}$) was obtained and the ethanol yield was considerably diminished (Fig 2 and Table 1). Compared to furfural, acetic acid was more toxic to the ZM4-hfq strain than ZM4-sigE, so a decrease in glucose consumption (40%), ethanol formation (50%) and growth (65%) was observed (Fig 2 and Table 1). In previous studies, different strategies such as chemical mutagenesis and Adaptive Laboratory Evolution (ALE) were used to generate resistant strains to acetate [11, 18]. *Z. mobilis* AcR (acetate-tolerant strain) is a mutant strain that capable to tolerate 20 $gL^{-1}$ acetate and efficiently produces ethanol [2]. Yang *et al.* (2010b) reported that the deletion of *hfq* gene in the AcR strain decreased acetate tolerance and concluded that the Hfq protein function and regulation is related to acetate resistance [18].

As it is observable in Fig 2D, when the concentration of acetic acid reached 5 $gL^{-1}$, ZM4-sigE exhibited better performance during ethanol fermentation in contrast to the other strains. Ethanol content was approximately two and four-fold higher than ZM4-hfq and parental strains, respectively (Table 1). Taken together, recombinant ZM4-sigE exhibited better growth ability as well as ethanol fermentation in both cases of inhibitors, furfural and acetic acid. Table 2 shows the previous studies on ethanol production performance concerning acetic acid stresses. According to this table, using different approaches, especially genetic engineering, had important achievements regarding *Z. mobilis* tolerance to the acetic acid and we introduced two recombinant engineered strains with significant tolerance to acetic acid. Since, multiple inhibitors' tolerance is a prerequisite for ethanol production from the lignocellulosic substrate, in the following step, this condition was evaluated.

## 3.3. Comparison of *Z. mobilis*, ZM4-sigE and ZM4-hfq growth and ethanol production in the presence of sugarcane bagasse hydrolysate

During the degradation of lignocellulosic materials, various toxic compounds with strong inhibitory effects on *Z. mobilis* cell growth are generated [3]. In the related studies, the inhibitory activity of hydrolysate compounds on the growth of *Z. mobilis* was strongly correlated to their hydrophobicity. Also, furfural, acetate and phenolic compounds are the major inhibitors in the hydrolysate [1, 3, 7, 18]. Although different physical, chemical, or biological approaches were used to detoxify these inhibitory compounds, it is not an economical industrial process. Therefore, the development of inhibitor-tolerant *Z. mobilis* strains is the main alternative strategy for economic lignocellulosic ethanol production [1, 3].

Sugarcane bagasse hydrolysate was fermented by wild type and recombinant strains. S2 Table shows the composition of hydrolysate and the results of fermentation experiments are presented in Fig 3. The presence of multiplex inhibitors in hydrolysate resulted in no substantial cell growth, sugar utilization and fermentation activity in the parental strain. Meanwhile, fermentation is supported <u>in</u> ZM4-hfq and ZM4-sigE strains. This observation implies that the mixture of inhibitors acted synergistically, as microorganism functionality is affected considerably compared to each inhibitor mentioned before [1]. In both recombinant strains, sugar remained unused in culture medium and the growth curve reached a plateau condition. Furthermore, the highest yield obtained in ZM4-hfq and ZM4-sigE strains were 0.21 and 0.36 $g_P$ $g_s^{-1}$, respectively (Fig 3 and Table 1).

This phenomenon advocated that the recombinant strains have a higher hydrolysate metabolic performance in comparison to the parental one. Additionally, these results are in

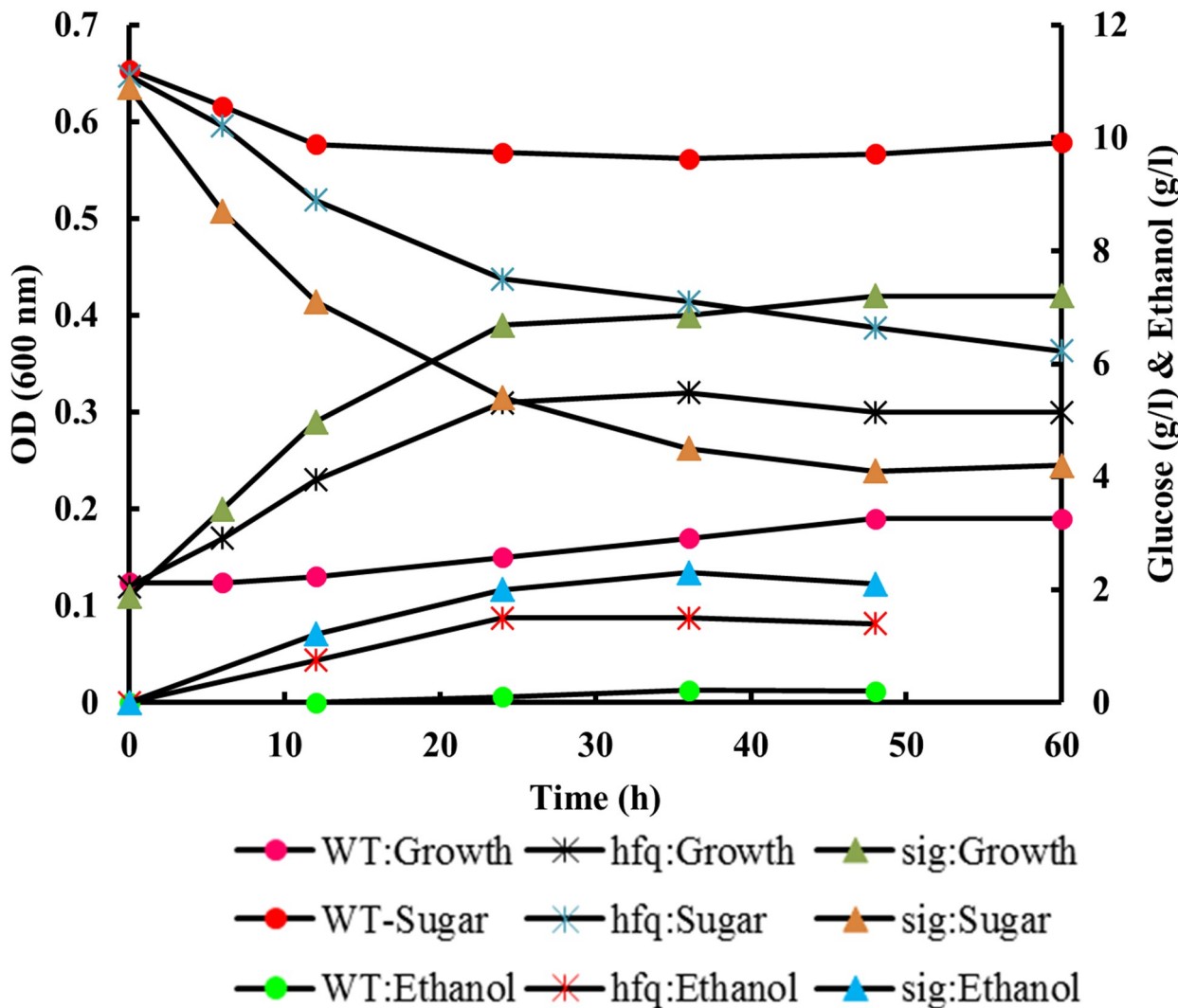

**Fig 3. Fermentation of the parental and recombinant strains in sugarcane bagasse hydrolysate.**

accordance with previous outcomes mentioned above, in which all strains were incubated in the presence of acetic acid or furfural individually, and the recombinants were superior to the others, with an emphasis on the ZM4-sigE strain. Table 3 demonstrates our results compared to other previous studies related to complex inhibitory compounds in the hydrolysate. According to this table, ethanol yield in hydrolysate medium has considerably reduced compared to the pure sugar [26–29]. Dong *et al.* 2013 and Zhao *et al.* 2014 reported that the ethanol production reduced 60–70% when *Z. mobilis* was grown in hydrolysate compared to glucose medium [26, 30]. These consequences verified the synergistic effects of inhibitory compounds in hydrolysate and showed that more efforts are needed to improve *Z. mobilis* cellular stress response.

The most reported studies were done with acetic acid-resistant strains such as pZB5 or 8b [27, 28, 31, 32] or an engineered *Z. mobilis* with xylose utilization ability [28, 29, 31, 33]. Our recombinant strains (ZM4-sigE and ZM4-hfq) cannot utilize pentose sugars like xylose in hydrolysate and are not adapted to high concentrations of acetic acid. However, the presented data showed that the ZM4-sigE and ZM4-hfq had acceptable activity and ethanol production performance in the presence of multiple hydrolysate inhibitory compounds.

**Table 3. Ethanol production by *Z. mobilis* strains in lignocellulosic hydrolysates.**

| *Z. mobilis* strain | Strain improvement | Carbon source | Inhibitors | Ethanol (gL$^{-1}$) | Yield | References |
|---|---|---|---|---|---|---|
| 8b | Engineered xylose utilizing | Corn stover | HMF (0.2), Acetic acid (5.9), Furfural (5.2) | 60 | 95% | [31] |
| ZM4 | Recombinant strain (NADPH-dependent alcohol dehydrogenase) | Corn stover | Furfural (0.62), Acetic acid (2.85), HMF (0.34) | 27 | 94% | [40] |
| 8b | Engineered xylose utilizing | Corn stover | Furfural (2.3), Acetic acid (16.3), HMF (3.7) | 48 | 70–75% | [27] |
| 8b | Engineered xylose utilizing | Corn stover | Acetic acid (11) | 36 | 85% | [32] |
| 2302 | Engineered xylose utilizing | Corn stover | Furfural (6.5), Syringaldehyde (1.3), HMF (0.1), Vanillin (21.5) | 39.1 | 80% | [41] |
| 2302 | Engineered xylose utilizing | Switch grass | Furfural (5.2), Vanillin (11.6), HMF (0.1), Syringaldehyde (1) | 38.3 | 81% | [41] |
| ZM401 | Flocculating *Z. mobilis* | Corn stover | Acetic acid (0.66), Furfural (0.49), HMF (0.28) | 27.5 | 46% | [26] |
| ZM4 | Recombinant strain (fdh) capable of degrading toxic inhibitor | Corn stover | HMF (0.29), Furfural (0.26), Formic acid (0.92) | 15 | 99% | [30] |
| TSH-01 | Engineered xylose utilizing strain | Corn stover | Furfural (0.6), Formic acid (1.2), Acetic acid (7.5) | 30 | 91–94% | [33] |
| ZM4 (pZB5) | Xylose utilization and tolerated acetic acid | Wheat straw | Furfural (0.67), HMF (0.08), Acetate (3.1), Levulinic acid (0.7), Formate (0.6) | 13.8 | 88% | [28] |
| ZM4 (pZB5) | Xylose utilization and tolerated acetic acid | Bagasse | Furfural (0.74), Acetate (4.1), HMF (0.1), Levulinic acid (0.8), Formate (0.7) | 17.7 | 84% | [28] |
| CP4 (pZB5) | Immobilized recombinant strain with Xylose utilization ability | Rice straw | - | 44.3 | 42–46% | [29] |
| ZM4 (pZB5) | Xylose utilization and tolerated acetic acid | Pine | Furfural (0.15), Formate (0.6), Acetate (1.8), HMF (0.14), Levulinic acid (0.4) | 0.2 | 25% | [28] |
| ZM4 (pZB5) | Xylose utilization and tolerated acetic acid | Sorghum straw | Acetate (2.3), Formate (1.2), Furfural (0.5), HMF (0.6), Levulinic acid (2.6) | 10.6 | 82% | [28] |
| ZM4 | Recombinant strain (*hfq* overexpression) | Sugar cane bagasse | Furfural (1.32), Acetic acid (3.61), HMF (0.8), phenolic compounds (0.78) | 1.5 | 44% | This work |
| ZM4 | Recombinant strain (*sig*E overexpression) | Sugar cane bagasse | Furfural (1.32), Acetic acid (3.61), HMF (0.8), phenolic compounds (0.78) | 3.6 | 72% | This wok |

## 3.4. Quantitative PCR (qRT-PCR) assay

Different genes for qRT-PCR assay were selected based on the main alcohol production genes (*pdc* and *adh*), the ability of recombinant strains to overexpress *sig*E and *hfq*, and the main genes related to furfural (*xylR*) and acetic acid (*nha*A) stresses according to previous studies [22, 34]. The gene expression profile was analyzed in wild type (Wt) *Z. mobilis*, recombinant strain (ZM4-sigE) and recombinant strain (ZM4-hfq) in untreated and treated (3 gL$^{-1}$ furfural or 5 gL$^{-1}$ acetic acid) conditions. As shown in Table 4, pyruvate decarboxylase (*pdc*) and alcohol dehydrogenase (*adh*) up-regulated in ZM4-sigE and ZM4-hfq compared to Wt. In the treated ZM4-sigE cells with 3 gL$^{-1}$ furfural and 5 gL$^{-1}$ acetic acid, *pdc* had 3 and 3.99-fold enhanced expression, respectively. These outcomes for *adh* were 2.48 and 2.73-fold, respectively. In the ZM4-hfq, the amount of heightened expression level was lower than ZM4-sigE. Growth data in Fig 2 supported these results in which Wt growth was strongly inhibited after treatment with high concentrations of furfural and acetic acid (Fig 2D and 2F).

Besides, the expression profile of *sig*E and *hfq* genes were examined in normal and stress conditions in ZM4-sigE and ZM4-hfq compared to Wt strain. As displayed in Table 4, *sig*E in recombinant ZM4-sigE exhibited 15.88-fold greater, and *hfq* in recombinant ZM4-hfq had 7.44-fold greater expression levels. These results indicated that overexpression of these two genes in *Z. mobilis* were successful. Based on the results, the expression level of *sig*E in

**Table 4. qRT-PCR results (avg±std, p<0.05).**

| Gene | Fold change | | | | | |
|---|---|---|---|---|---|---|
| | Without stress | | Furfural stress (3 gL⁻¹) | | Acetic acid stress (5 gL⁻¹) | |
| | *sig*E/ Wt | *hfq*/ Wt | *sig*E/ Wt | *hfq*/ Wt | *sig*E/ Wt | *hfq*/ Wt |
| *pdc* | +1.17±0.14 | +1.03±0.09 | +3.05±0.2 | +2.01±0.05 | +3.99±0.3 | +2.69±0.05 |
| *ahd* | - 0.82±0.08 | +1.23±0.06 | +2.48±0.3 | +1.99±0.1 | +2.73±0.09 | +1.94±0.1 |
| *sig*E | +15.88±0.9 | +1.14±0.1 | +7.45±0.72 | -1.37±0.3 | +6.59±0.7 | -2.13±0.3 |
| *hfq* | - 0.77±0.07 | +8.11±0.5 | -1.55±0.06 | +5.36±0.8 | -1.71±0.09 | +4.44±0.24 |
| *xylR* | - 1.57±0.07 | - 1.39±0.07 | +3.89±0.23 | +2.73±0.14 | +2.03±0.15 | +2.22±0.16 |
| *nha* A | +1.45±0.16 | +1.01±0.07 | +1.7±0.12 | +2.16±0.2 | +4.46±0.18 | +3.13±0.21 |

(+ and - signs refer to gene up and down-regulation, respectively)

recombinant ZM4-sigE was more successful than recombinant ZM4-hfq (Table 4). One possible reason for the lower expression level of *hfq* gene could be the presence of regulatory RNA regions within a transcript, particularly in the 5′ untranslated region (5′UTR) for *hfq* gene [19]. Based on the Cho *et al.* (2017) study, this regulatory RNA controls the expression levels of *hfq* gene in response to different environmental and stress condition [19]. This phenomenon may be an acceptable reason to why recombinant ZM4-hfq is weaker than recombinant ZM4-sigE in cells treated with inhibitory compounds. Recently, Benoliel *et al.* (2019) overexpressed *sig*E in *Z. mobilis* and their qPCR results showed 169-fold enhanced expression compared to the control. They reported that this high amount of SigE protein is significantly associated with the improvement in tolerance to high-temperature conditions but there is no clear evidence that this protein protects *Z. mobilis* in ethanol stress [15].

In the furfural stress response, the expression level of *sig*E in ZM4-sigE and *hfq* in ZM4-hfq strains were 7.44 and 5.36-fold up-regulated compared to Wt, respectively. While the average expression level of *hfq* in ZM4-sigE and *sig*E in ZM4-hfq down-regulated by -1.37 and -1.55 fold, respectively. These results support the hypothesis that the activity of these two regulatory proteins overlap. In acetic acid stress conditions, the expression plans of *sig*E and *hfq* genes showed similar behaviors. Zhang *et al.* (2010) overexpressed the regulatory protein IrrE from *Deinococcus radiodurans* in *Z. mobilis*. They revealed that the expression level of *sig*E enhanced by 8.2 and 6-fold at pH 3.8 and in 12% ethanol, respectively. Our findings on furfural, acetic acid and multiple inhibitor compounds in sugarcane bagasse hydrolysate, along with Zhang *et al.* (2010) outcomes on low pH and ethanol stress and Benoliel *et al.* (2019) on high temperature, confirmed Seo's theory (2005) about the critical role of *sig*E in the response to various stress conditions [15, 20, 35].

Tan *et al.* 2015 showed that in the furfural stress, the expression level of *xylR* aldo/keto reductase (ZMO0976) had increased 15.83-fold in one of the RpoD (σ⁷⁰) mutants achieved from error-prone PCR libraries, compared to control cells [22]. Our results indicated that the expression of *xylR* in recombinant strains ZM4-sigE and ZM4-hfq was increased 3.89 and 2.73-fold in cells treated with 3 gL⁻¹ furfural compared to Wt, respectively. We also found an increase in the expression of this gene in acetic acid stress so this gene is probably responsible for a typical response in furfural and acetic acid stresses. Agrawal and Chen (2011) presented that *xylR* is related to converting xylose to xylitol and has 150-fold higher affinity to benzaldehyde than xylose. Furthermore, their findings together with Tan *et al.* (2015) study demonstrated that the *xylR* could mitigate the furfural toxicity and play a critical role in the tolerance of furfural in *Z. mobilis* [22, 36]. Similar to these studies, our results emphasized that the

overexpression of transcriptional regulatory factors, *hfq* and *sig*E led to the increase of *xylR* expression with respect to the control cells (Table 4).

Transcriptomic and proteomic analysis of acetic acid-resistant strains of *Z. mobilis*, AcR (acetic acid resistant), showed that the sodium-proton antiporter gene *nha*A (ZMO0117) plays a critical role in acetic acid resistance [13, 34]. Our results demonstrated than in recombinant strains ZM4-sigE and ZM4-hfq with a high content of regulatory factors SigE and Hfq protein, the expression level of *nha*A was increased 4.4 and 3.13-fold, respectively. This transcriptional activation was carried out after acetic acid (5 gL$^{-1}$) treatment compared to control. Moreover, Yang *et al.* (2010a) showed that the expression level of *nha*A in AcR was 16-fold higher than the *Z.mobilis* ZM4 by microarray analysis [13]. This finding emphasized this theory that SigE and Hfq proteins are important regulatory factors for *nha*A expression.

Totally, in the furfural stress, the expression level of *xylR* gene was higher than *nha*A in both recombinant ZM4-sigE and ZM4-hfq. Besides, in the acetic acid treatment; *nha*A gene expression was higher than *xylR*. These findings emphasize the important role of *xylR* and *nha*A genes in tolerance to furfural and acetic acid, respectively and proposed that the control of these genes are related to these two regulatory factors. In addition, as mentioned earlier, the lower expression level of *hfq* gene can be attributed to the presence of regulatory RNA regions within a transcript, particularly in the 5′ untranslated region (5′UTR) of the *hfq* gene [19]. This regulatory RNA supports this hypothesis as to why the expression levels of xylR and nhaA were different between ZM4-hfq and ZM4-sigE while also having better function of recombinant ZM4-sigE compared to ZM4-hfq in tolerance to lignocellulosic inhibitors.

## Conclusion

The tolerance of a microorganism toward stresses is one of the crucial challenges for cost-competitive bioethanol production from lignocellulosic substrates. Due to the overlap of cell stress responses, manipulation of the global regulatory landscape may show different effects on various aspects of bacterial metabolism. In this study, the relationship between the upregulation of the regulatory elements, *sig*E and *hfq* genes, confirmed cell phenotypic improvement in response to a single (furfural or acetic acid) and multiple (sugarcane bagasse hydrolysate) inhibitors. As revealed in our fermentation and qRT-PCR results, up-regulated *sig*E gene is more efficient in both conditions mentioned above and poses high levels of tolerance as well as superior growth and fermentation performance. As recombinant *Z. mobilis* strains showed the efficiency of the biomass conversion to ethanol, this would be beneficial to exhibit better fitness under the toxicant-containing environment. The entire outcomes led us to conclude that strain engineering at the level of transcription regulation improves the efficacy of an organism in response to various stresses and can be used for other industrially important microorganisms.

## Supporting information

**S1 Table. Primers used for the preparation of recombinant plasmids and qPCR.**
(DOCX)

**S2 Table. Some of the sugarcane bagasse hydrolysate ingredients.**
(DOCX)

**S1 Fig. pBBR1MCS-2-sigE and pBBR1MCS-2-hfq plasmid construction.**
(TIF)

**S1 Graphical abstract. Schematic design of genetic engineering and fermentation process of recombinant *Z. mobilis*.**
(TIF)

## Acknowledgments

Special thanks to Prof. Mehrdad Azin for his help to provide some chemicals.

## Author Contributions

**Conceptualization:** Hoda Nouri.

**Funding acquisition:** Hamid Moghimi.

**Methodology:** Hoda Nouri.

**Project administration:** Hamid Moghimi.

**Supervision:** Hamid Moghimi, Sayed-Amir Marashi, Elahe Elahi.

**Visualization:** Hoda Nouri.

**Writing – original draft:** Hoda Nouri.

**Writing – review & editing:** Hoda Nouri, Hamid Moghimi, Sayed-Amir Marashi.

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
