## [Decision Letter · Decision Letter 0]

15 Jun 2020

PONE-D-20-14531

Impact of hfq and sigE on the tolerance of Zymomonas mobilis ZM4 to furfural and acetic acid stresses

PLOS ONE

Dear Dr. Moghimi,

Thank you for submitting your manuscript to PLOS ONE. After careful consideration, we feel that it has merit but does not fully meet PLOS ONE’s publication criteria as it currently stands. Therefore, we invite you to submit a revised version of the manuscript that addresses the points raised during the review process.

Take special care to properly discuss the results in the revised manuscript. Moreover, you should properly address the comment from Reviewer 1 and point out what new knowledge your work has generated and how the approach here is different from what is already been published.

We look forward to receiving your revised manuscript.

Kind regards,

Leonidas Matsakas

Academic Editor

PLOS ONE

Journal Requirements:

"The University of Tehran partially supported this work."

3.Thank you for stating in your Acknowledgments Section:

""The University of Tehran partially supported this work."

4. Please ensure that you refer to "Figure Graphical Abstract" in your text as, if accepted, production will need this reference to link the reader to the figure.

Reviewers' comments:

Reviewer's Responses to Questions

**Comments to the Author**

1. Is the manuscript technically sound, and do the data support the conclusions?

Reviewer #1: Yes

Reviewer #2: Yes

2. Has the statistical analysis been performed appropriately and rigorously? 

Reviewer #1: Yes

Reviewer #2: Yes

3. Have the authors made all data underlying the findings in their manuscript fully available?

Reviewer #1: Yes

Reviewer #2: Yes

4. Is the manuscript presented in an intelligible fashion and written in standard English?

Reviewer #1: Yes

Reviewer #2: Yes

5. Review Comments to the Author

Reviewer #1: Comments to PONE-D-20-14531

The authors constructed two stress tolerant Zymomonas mobilis strains by introducing hfq and sigmaE. The topic sounds interesting. However, the research lacks innovation.

Actually, Yang et al has used this approach to construct stress tolerant strain by overexpression hfq in Z. mobilis (Shihui Yang, Dale A Pelletier,Tse-Yuan S Lu, and Steven D Brown. The Zymomonas mobilis regulator hfq contributes to tolerance against multiple lignocellulosic pretreatment inhibitors. BMC Microbiol, 2010; 10: 135). In another research, Tiago et al also used different sigma factor to construct stress tolerant Z. mobilis strains (Tiago Benoliel, Marciano Régis Rubini, Carolina de Souza Baptistello, Christiane Ribeiro Janner, Vanessa Rodrigues Vieira, Fernando Araripe Torres, Adrian Walmsley & Lidia Maria Pepe de Moraes. Physiological effects of overexpressed sigma factors on fermentative stress response of Zymomonas mobilis. Brazilian Journal of Microbiology, 2020, 51: 65-75)

Reviewer #2: This study evaluated the effect of two transcriptional elements, hfq and sigE, on stress tolerance of Z. mobilis to furfural, acetic acid and sugarcane hydrolysate. It demonstrated the role of hfq and sigE on resistance to toxic byproducts from pretreatment. Although this manuscript presented a complete study, there are many concerns should be addressed.

1. The authors need to discuss and dig deeper in the results section rather than simply listing the trends of the diagram without extended analysis. There are a lot of interesting things to talk about, for example, in Table 3, why the expression of xylR and nhaA was significantly different between ZM4-hfq and ZM4-sigE. What are the potential regulation of hfq and sigE on expression of xylR or nhaA?

2. Line 23-25, please rewrite these sentences to ensure that they are rational.

3. Line 27-29, please revise this sentence.

4. Line 49-51, provide references.

5. Line 69, please reform the reference.

6. Line 144, please replace five with 5.

7. Line 193-194, please make sure that this sentence is correct.

8. Figure 3 and Figure 4. Please adjust the color of line and shape of dots to make them clear.

9. Figure 3a. Please explain why the glucose will slightly increase after reducing to 0 g/L.

10. Table 3, the author wrote “Avg±std” in the title, but no such format number was showed in this table.

11. The 1st and 2nd paragraph in “3. Results and Discussion” should be put into “1. Introduction”.

12. Figure 1 should be in supplementary materials

13. Table S3 should be in the main text. Because the authors cited it many times. It is important.

14. The language and sentence should be polished carefully.

6. PLOS authors have the option to publish the peer review history of their article (what does this mean?). If published, this will include your full peer review and any attached files.

Reviewer #1: No

Reviewer #2: Yes: Chen-Guang Liu

---

## [Author Response · Author response to Decision Letter 0]

21 Aug 2020

Journal Requirements:

 -Thanks to the editor for the comment, the manuscript format has been corrected according to PLOS ONE's style requirements.

"The University of Tehran partially supported this work." We note that you have provided funding information that is not currently declared in your Funding Statement. However, funding information should not appear in the Acknowledgments section or other areas of your manuscript. We will only publish funding information present in the Funding Statement section of the online submission form.

Please remove any funding-related text from the manuscript and let us know how you would like to update your Funding Statement. Currently, your Funding Statement reads as follows: "The author(s) received no specific funding for this work."

 -Thanks to the editor for the comment, Acknowledgments Section was edited. 

3. Please ensure that you refer to "Figure Graphical Abstract" in your text as, if accepted, production will need this reference to link the reader to the figure.

 -Thanks to the editor for the comment, Graphical Abstract has been cited in the manuscript (line 115)

-Supporting information legends were added at the end of the manuscript and correctly cited in the text. 

5. Review Comments to the Author

Reviewer #1: Comments to PONE-D-20-14531

The authors constructed two stress tolerant Zymomonas mobilis strains by introducing hfq and sigmaE. The topic sounds interesting. However, the research lacks innovation.

Q1: Actually, Yang et al has used this approach to construct stress tolerant strain by overexpression hfq in Z. mobilis (Shihui Yang, Dale A Pelletier,Tse-Yuan S Lu, and Steven D Brown. The Zymomonas mobilis regulator hfq contributes to tolerance against multiple lignocellulosic pretreatment inhibitors. BMC Microbiol, 2010; 10: 135). In another research, Tiago et al also used different sigma factor to construct stress tolerant Z. mobilis strains (Tiago Benoliel, Marciano Régis Rubini, Carolina de Souza Baptistello, Christiane Ribeiro Janner, Vanessa Rodrigues Vieira, Fernando Araripe Torres, Adrian Walmsley & Lidia Maria Pepe de Moraes. Physiological effects of overexpressed sigma factors on fermentative stress response of Zymomonas mobilis. Brazilian Journal of Microbiology, 2020, 51: 65-75)

A1: In the study of Yang et al. developed a Z. mobilis hfq insertion mutant. The resulting strain designated as AcRIM0347 and its role was investigated in model lignocellulosic inhibitors including acetate, vanillin, furfural and hydroxymethylfurfural. Also, they named pBBR3DEST42 for hfq expression (p42-0347). According to their results, only Z. mobilis hfq mutant (AcRIM0347) was assayed in the presence of model lignocellulosic pretreatment inhibitors. In our study, the overexpression of hfq in ZM4 strain and its behaviour in furfural, acetic acid and multi-complex inhibitors was investigated. Besides, in the Yang study, there is no evidence about ethanol production and their focus was mainly on the growth of Z mobilis.

Moreover, the recombinant hfq expression profile in the presence of multi-complex of lignocellulosic inhibitors was not analyzed. In our study, for the first time, the relation between sugar consumption, ethanol and biomass production were investigated in wild type and recombinant hfq Z. mobilis in the high and low concentrations of lignocellulosic inhibitors. Finally, the behaviour of recombinant strain was studied in real sugarcane hydrolysate with complex inhibitors and the expression level of some indicator genes were compared in recombinant strain. We believe our study affords more insight about the hfq as a multifunctional global regulatory factor. 

In the study of Benoliel et al. they overexpressed sigma 32 and sigma 24 (Sig E) in Z. mobilis and their influence were tested in response to osmotic and ethanol stresses. Here, we studied the overexpression of sigma 24 (Sig E) in response to the lignocellulosic inhibitors stresses for the first time. According to Benoliel et al. results, it was not possible to conclude if sigma 24 affects ethanol tolerance in Z. mobilis, but the overexpression of this sigma factor led to a decrease in ethanol productivity. To our knowledge, our study is the first report of Sig E overexpression in Z. mobilis that has evaluated the recombinant strains on the lignocellulosic inhibitors such as furfural and acetic acid. Based on our results, in comparison to Benoliel et al. study, Sig E overproduction has significant effects on cell tolerance to the lignocellulosic inhibitors and improves ethanol production in comparison to wild type strain. 

We believed that our detailed experimental achievements about two transcriptional elements help to gain deep insights in Z. mobilis stress response mechanisms.

Respect to the reviewers' comment, the novelty of the study was more highlighted in the manuscript (line 86-96). 

Reviewer #2: This study evaluated the effect of two transcriptional elements, hfq and sigE, on stress tolerance of Z. mobilis to furfural, acetic acid and sugarcane hydrolysate. It demonstrated the role of hfq and sigE on resistance to toxic byproducts from pretreatment. Although this manuscript presented a complete study, there are many concerns should be addressed.

Thanks to the honorable reviewer of the article

Q1. The authors need to discuss and dig deeper in the results section rather than simply listing the trends of the diagram without extended analysis. There are a lot of interesting things to talk about, for example, in Table 3, why the expression of xylR and nhaA was significantly different between ZM4-hfq and ZM4-sigE. What are the potential regulation of hfq and sigE on expression of xylR or nhaA?

A1. We thank the reviewer for his/her constructive comment. Accordingly, to improve our discussion, we added the following paragraphs to the manuscript (lines 231-237, 243-247, 261-267, 269-271, 282-287, 297-299, 304-311, 329-334, 362-369, 408-417).

Q2. Line 23-25, please rewrite these sentences to ensure that they are rational.

A2. The mentioned sentences rewrote to clarify the meaning (lines 32-34).

Q3. Line 27-29, please revise this sentence. 

A3. The mentioned lines revised (lines 35-38).

Q4. Line 49-51, provide references. 

A4. References were added to the reference list.

Q5. Line 69, please reform the reference. 

A5. The mentioned sentence was checked.

Q6. Line 144, please replace five with 5. 

A6. A replacement was made.

Q7. Line 193-194, please make sure that this sentence is correct.

A7. The mentioned sentence was checked.

Q8. Figure 3 and Figure 4. Please adjust the color of line and shape of dots to make them clear. 

A8. Based on the reviewer's comment, some changes were made in figures.

Q9. Figure 3a. Please explain why the glucose will slightly increase after reducing to 0 g/L. 

A9. Prolonged starvation and nutrient deprivation will result in a drastic decline in optical density followed by cell death. As a consequence of the cell lysis, cellular contents as nutrients are released that is attributed to a slight increase in glucose content. The rest of the population scavenge these nutrients as carbon and nitrogen sources.

• Juana María Navarro Llorens, Antonio Tormo, Esteban Martínez-García, "Stationary phase in gram-negative bacteria", FEMS Microbiology Reviews, Volume 34, Issue 4, July 2010, Pages 476–495.

• Jaishankar, Jananee, and Preeti Srivastava. "Molecular basis of stationary phase survival and applications." Frontiers in microbiology 8 (2017): 2000.

Q10. Table 3, the author wrote "Avg±std" in the title, but no such format number was showed in this table.

A10. Thank the reviewer for precise comment, standard deviations for all quantifications were added in the manuscript.

Q11. The 1st and 2nd paragraph in "3. Results and Discussion" should be put into "1. Introduction". 

A11. The mentioned paragraphs re-organized and put into the introduction section.

Q12. Figure 1 should be in supplementary materials

A12. According to the reviewer's comment, a replacement was done.

Q13. Table S3 should be in the main text. Because the authors cited it many times. It is important.

A13. Table S3 was added to the main text of the manuscript.

Q14. The language and sentence should be polished carefully. 

A14. We thank the reviewer for this comment. A native speaker edited the manuscript.

---

## [Decision Letter · Decision Letter 1]

14 Sep 2020

PONE-D-20-14531R1

Impact of hfq and sigE on the tolerance of Zymomonas mobilis ZM4 to furfural and acetic acid stresses

PLOS ONE

Dear Dr. Moghimi,

Thank you for submitting your manuscript to PLOS ONE. After careful consideration, we feel that it has merit but does not fully meet PLOS ONE’s publication criteria as it currently stands. Therefore, we invite you to submit a revised version of the manuscript that addresses the points raised during the review process.

We look forward to receiving your revised manuscript.

Kind regards,

Leonidas Matsakas

Academic Editor

PLOS ONE

Reviewers' comments:

Reviewer's Responses to Questions

**Comments to the Author**

1. If the authors have adequately addressed your comments raised in a previous round of review and you feel that this manuscript is now acceptable for publication, you may indicate that here to bypass the “Comments to the Author” section, enter your conflict of interest statement in the “Confidential to Editor” section, and submit your "Accept" recommendation.

Reviewer #2: (No Response)

2. Is the manuscript technically sound, and do the data support the conclusions?

Reviewer #2: Yes

3. Has the statistical analysis been performed appropriately and rigorously? 

Reviewer #2: Yes

4. Have the authors made all data underlying the findings in their manuscript fully available?

Reviewer #2: No

5. Is the manuscript presented in an intelligible fashion and written in standard English?

Reviewer #2: Yes

6. Review Comments to the Author

Reviewer #2: I consider that requirements from previous peer round were conveniently addressed. Based on this aspect, the manuscirpt can be accepted. However, I still have some comments for the improvement of this manucript.

1. Q7. Line 193-194, please make sure that this sentence is correct.

A7. The mentioned sentence was checked.

The sentence has not been corrected.

2. Q8. Figure 3 and Figure 4. Please adjust the color of line and shape of dots to make them clear.

A8. Based on the reviewer's comment, some changes were made in figures.

The color of line and shape of dots should be standardized based on the names of gene and Y-axis

7. PLOS authors have the option to publish the peer review history of their article (what does this mean?). If published, this will include your full peer review and any attached files.

Reviewer #2: **Yes: **Chen-Guang Liu

---

## [Author Response · Author response to Decision Letter 1]

22 Sep 2020

22 September 2020 

Dear Prof. Heber

Editor in Chief of PLOS ONE

We are very excited to have been allowed to revise our manuscript "Impact of hfq and sigE on the tolerance of Zymomonas mobilis ZM4 to furfural and acetic acid stresses" (Manuscript Number: PONE-D-20-14531R1). Thanks for the advice, which will help us improve it to a better scientific level. We have revised the manuscript according to the reviewer comments and the proposed changes have taken place. 

Yours Sincerely

Hamid Moghimi

Email: hmoghimi@ut.ac.ir

PONE-D-20-14531R1

Impact of hfq and sigE on the tolerance of Zymomonas mobilis ZM4 to furfural and acetic acid stresses

PLOS ONE

Dear Dr. Moghimi,

Thank you for submitting your manuscript to PLOS ONE. After careful consideration, we feel that it has merit but does not fully meet PLOS ONE’s publication criteria as it currently stands. Therefore, we invite you to submit a revised version of the manuscript that addresses the points raised during the review process.

6. Review Comments to the Author

Reviewer #2: I consider that requirements from previous peer round were conveniently addressed. Based on this aspect, the manuscirpt can be accepted. However, I still have some comments for the improvement of this manucript.

1. Q7. Line 193-194, please make sure that this sentence is correct.

A7. The mentioned sentence was checked. The sentence has not been corrected.

Thanks the reviewer for the comment, the paragraph was edited according to the reviewer comment. 

“As we expected, when no inhibitor was added to the culture medium, the parental and both recombinant strains showed similar growth pattern. These observations confirmed that the growth ability of ZM4-sigE and ZM4-hfq were not influenced during genetic engineering procedures and remained constant like control sample”. (Lines 200-203)

2. Q8. Figure 2 and Figure 3. Please adjust the color of line and shape of dots to make them clear.

A8. Based on the reviewer's comment, some changes were made in figures.

The color of line and shape of dots should be standardized based on the names of gene and Y-axis

Thanks the reviewer for the comment, the Fig 2 and 3 was corrected and same symbols used base on the names of gene and Y-axis.

---

## [Editor Report · Decision Letter 2]

24 Sep 2020

Impact of hfq and sigE on the tolerance of Zymomonas mobilis ZM4 to furfural and acetic acid stresses

PONE-D-20-14531R2

Dear Dr. Moghimi,

We’re pleased to inform you that your manuscript has been judged scientifically suitable for publication and will be formally accepted for publication once it meets all outstanding technical requirements.

Kind regards,

Leonidas Matsakas

Academic Editor

PLOS ONE
---

## [Editor Report · Acceptance letter]

29 Sep 2020

PONE-D-20-14531R2 

Impact of *hfq* and *sig*E on the tolerance of *Zymomonas*
*mobilis* ZM4 to furfural and acetic acid stresses 

Dear Dr. Moghimi:

I'm pleased to inform you that your manuscript has been deemed suitable for publication in PLOS ONE. Congratulations! Your manuscript is now with our production department. 

Kind regards, 

on behalf of

Dr. Leonidas Matsakas 

Academic Editor

PLOS ONE